# Recessive pathogenic variants in *MCAT* cause combined oxidative phosphorylation deficiency

Bryn D Webb[1,2,3,4]*†, Sara M Nowinski[5]†, Ashley Solmonson[6], Jaya Ganesh[2], Richard J Rodenburg[7], Joao Leandro[2], Anthony Evans[2], Hieu S Vu[6], Thomas P Naidich[8], Bruce D Gelb[2,3,4], Ralph J DeBerardinis[6,9], Jared Rutter[9,10], Sander M Houten[2]

[1]Department of Pediatrics and Center for Human Genomics and Precision Medicine, University of Wisconsin School of Medicine and Public Health, Madison, WI, United States; [2]Department of Genetics and Genomic Sciences, Icahn School of Medicine at Mount Sinai, New York, NY, United States; [3]Department of Pediatrics, Icahn School of Medicine at Mount Sinai, New York, NY, United States; [4]Mindich Child Health and Development Institute, Icahn School of Medicine at Mount Sinai, New York, NY, United States; [5]Department of Metabolism and Nutritional Programming, Van Andel Institute, Grand Rapids, MI, United States; [6]Children's Medical Center Research Institute, University of Texas Southwestern Medical Center, Dallas, TX, United States; [7]Department of Pediatrics, Nijmegen Center for Mitochondrial Disorders, Radboud University Medical Center, Nijmegen, Netherlands; [8]Department of Radiology, Icahn School of Medicine at Mount Sinai, New York, NY, United States; [9]Howard Hughes Medical Institute, Chevy Chase, MD, United States; [10]Department of Biochemistry, University of Utah, Salt Lake City, UT, United States

*For correspondence:
bdwebb@wisc.edu

†These authors contributed equally to this work

Competing interest: The authors declare that no competing interests exist.

**Abstract** Malonyl-CoA-acyl carrier protein transacylase (MCAT) is an enzyme involved in mitochondrial fatty acid synthesis (mtFAS) and catalyzes the transfer of the malonyl moiety of malonyl-CoA to the mitochondrial acyl carrier protein (ACP). Previously, we showed that loss-of-function of mtFAS genes, including *Mcat*, is associated with severe loss of electron transport chain (ETC) complexes in mouse immortalized skeletal myoblasts (Nowinski et al., 2020). Here, we report a proband presenting with hypotonia, failure to thrive, nystagmus, and abnormal brain MRI findings. Using whole exome sequencing, we identified biallelic variants in *MCAT*. Protein levels for NDUFB8 and COXII, subunits of complex I and IV respectively, were markedly reduced in lymphoblasts and fibroblasts, as well as SDHB for complex II in fibroblasts. ETC enzyme activities were decreased in parallel. Re-expression of wild-type *MCAT* rescued the phenotype in patient fibroblasts. This is the first report of a patient with *MCAT* pathogenic variants and combined oxidative phosphorylation deficiency.

## Editor's evaluation

This paper is a valuable report of a human subject with an MCAT mutation showing reduced mitochondrial activity. Analysis of cells from this subject convincingly showed similar abnormalities as previously demonstrated for MCAT deficiency in the mouse. This work adds to the field of mitochondrial disease genes and their impact on human physiology.

## Introduction

Malonyl-CoA-acyl carrier protein transacylase (MCAT) is a catalytic component of the type II malonyl-CoA-dependent system for fatty acid synthesis (FAS) in the mitochondria. MCAT acts at the second step of mitochondrial fatty acid synthesis (mtFAS) whereby malonyl-CoA is used to generate malonyl-acyl carrier protein (ACP) (*Hiltunen et al., 2009*).

Octanoate, the precursor for the co-factor lipoate, is thought to be the main product of the mtFAS system. Lipoylation is necessary for the activity of the pyruvate dehydrogenase (PDHc), 2-oxoglutarate dehydrogenase (OGDHc), branched-chain ketoacid dehydrogenase (BCKDHc), and 2-oxoadipate dehydrogenase complexes, and the glycine cleavage system (*Solmonson and DeBerardinis, 2018*).

A mouse model with reduced *Mcat* expression displays decreased physical activity, reduced muscle strength, and premature aging with a reduced life span. Additionally, these animals have elevated skeletal muscle lactate, blood lactate, and ketone bodies, and have lipoylation defects in OGDHc, PDHc, and BCKDHc. The authors postulated that the mouse phenotype is due primarily to decreased generation of octanoate, leading to decreased lipoylation of PDHc and OGDHc and decreased flux through the citric acid cycle with subsequent disruption of energy metabolism (*Smith et al., 2012*).

Although MCAT has not been demonstrated to have a role in oxidative phosphorylation, the mitochondrial ACP encoded by *NDUFAB1* has been shown to be essential for complex I assembly (*Stroud et al., 2016*). In an effort to further study the mtFAS pathway, Nowinski et al. recently generated hypomorphic *Mcat* clones in an immortalized mouse skeletal myoblast line using CRISPR-Cas9. These hypomorphic *Mcat* clones have a 30–40% decreased basal respiration rate compared to controls, which is explained by markedly decreased levels of fully assembled complexes I, II, and IV. Metabolomics and carbon-labeling studies are all consistent with decreased function of the TCA cycle and oxidative phosphorylation in the hypomorphic *Mcat* mutant cells (*Nowinski et al., 2020*). This work demonstrates that the role of mtFAS extends beyond protein lipoylation and includes a crucial role in the maintenance of electron transport chain (ETC) activity. We now present a proband with mitochondrial disease and a biallelic missense variant in *MCAT*.

## Case presentation

A 2-year-old boy, born prematurely, with a past medical history notable for failure to thrive and global developmental delay presented for genetics evaluation. Birth weight was 1.106 kg. The patient was hospitalized in the neonatal intensive care unit for respiratory support (oxygen via nasal cannula) and was discharged home at 7 weeks. The patient was being followed by ophthalmology for retinopathy of prematurity and at 8 months of age was noted to have esotropia and horizontal jerk nystagmus that decreased with convergence. He was prescribed eye patching resulting in some improvement of the esotropia, but at 19 months was noted to have new-onset vertical nystagmus. A brain MRI completed at 21 months was notable for bilaterally symmetrical increase in signal intensity on T2- and T2 FLAIR (fluid attenuated inversion recovery)-weighted sequences in the dorsal medulla, periaqueductal gray matter, substantia nigra, medial hypothalamus, and habenular nuclei. There were two asymmetric foci in the subcortical white matter of the cerebellar hemispheres. There was no surrounding edema and no abnormal contrast enhancement. Developmentally, at 2 years of age the patient could say approximately two to three words, had no two-word phrases, and could not walk independently. On exam, his head circumference was at the 0.4 percentile ($Z=-2.59$), height was at the 0.46 percentile ($Z=-2.61$), and weight was at the <0.01 percentile ($Z=-3.73$). He had plagiocephaly with sagittal suture prominence, but was otherwise nondysmorphic. Vertical and horizontal nystagmus were present. Neurological exam was notable for truncal and peripheral hypotonia, decreased muscle mass, and brisk reflexes. Serum creatine kinase, blood lactate, pyruvate, ammonia, acylcarnitine profile, plasma amino acids, and urine organic acids were normal. Repeat brain MRI at 3 years 3 months showed substantially increased signal intensity and extent of involvement at the same sites, with no edema, no mass effect, and no abnormal contrast enhancement at any site (*Figure 1A–F*; *Naidich and Duvernoy, 2009*; *Standring, 2020*). No new sites of involvement were identified. The remaining portions of the brain showed no abnormality. Brain MR spectroscopy at short TE was normal within the right thalamus and left subcortical white matter. Chromosomal microarray revealed no pathogenic copy number variation, but absence of heterozygosity of unknown clinical significance totaling 32.2 Mb was detected. Family history was significant for an older sibling born with cleft lip, who died at a year of age from a choking episode. Visual symptoms were not seen in this sibling. The proband has two younger siblings

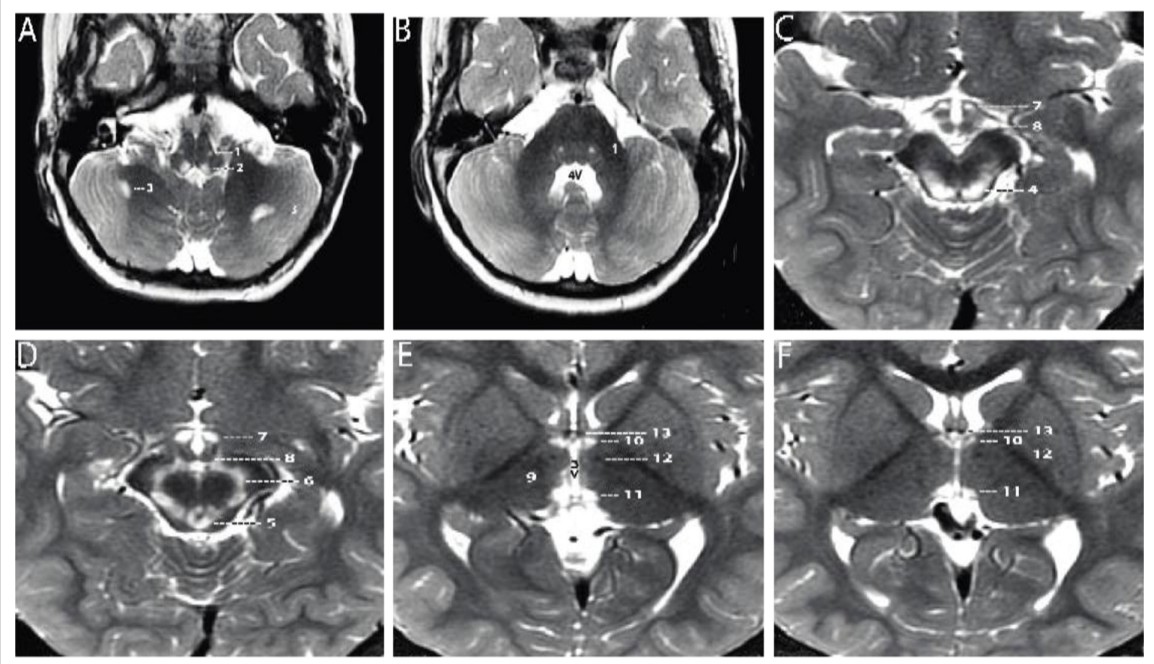

**Figure 1.** Axial T2-weighted MRI sections of the brain displayed from inferior to superior. (**A–B**) Sections through the brainstem and cerebellum. Sections through the medulla (**A**) and mid pons (**B**) demonstrate well-defined, symmetrical signal increase in the central tegmental tracts (1) and the nuclei prepositus hypoglossi (2) with very limited, slightly asymmetrical involvement of the white matter of both cerebellar hemispheres (3). 4V=fourth ventricle. (**C–D**) Sections through the midbrain display intense symmetrical increase in signal in the tectum (4), peri-aqueductal gray matter (5), substantia nigra (6), and medial hypothalamus (7). The mammillary bodies (8) are spared. (**E–F**) Sections through the hypothalamus (**E**) and thalamus (**F**) display intense symmetrical increase in signal in the medial thalamus (9), anteromedial thalamus (10), and habenular nuclei (11), with sparing of the low signal mammillothalamic tracts (12) and anterior columns of the fornix (13). No abnormal signal was identified in any other portions of the brain. 3V=third ventricle.

who are healthy and have normal development. Family history was otherwise non-contributory, and while both parents are from the same region of Mexico, consanguinity was denied.

## Results

By whole exome sequencing, we identified a homozygous variant in *MCAT* c.812T>C; p.T271I (rs760294168; NM_173467.4) in the proband, which was heterozygous in both parents consistent with autosomal recessive inheritance. This variant had a high CADD score of 28.9, is predicted to be damaging and probably damaging by SIFT and PolyPhen-2, respectively, and is in gnomAD at very low frequency (2/251,156 alleles; no homozygotes reported). In addition, p.T271 is highly conserved across species, from human to *Xenopus* and zebrafish.

To evaluate whether this patient's symptoms may be due to MCAT deficiency with disrupted mtFAS and mitochondrial pathology, we first performed immunoblot analysis in lymphoblast and fibroblast samples from the proband and controls. In lymphoblasts, immunoblot analysis revealed a decreased protein level for MCAT as well as NDUFB8 and COXII, which are selected subunits of complex I and IV, respectively (*Figure 2A–B*, *Figure 2—figure supplement 1*). In fibroblasts, immunoblot analysis revealed decreased protein levels for NDUFB8, SDHB, UQCRC2, and COXII which are selected subunits of complexes I, II, III, and IV, respectively (*Figure 2D*, *Figure 2—figure supplement 1*). Respiratory chain enzyme activities were evaluated in the proband's fibroblasts and revealed decreased activities for complexes I, II, and IV (*Table 1*; *Rodenburg, 2011*).

In mammals, ETC components assemble into respiratory supercomplexes composed of complexes I, III, and IV in varying stoichiometries (*Letts et al., 2016*). Intact, fully assembled complex I-containing supercomplexes and complex IV-containing supercomplexes were decreased in abundance in fibroblasts from the patient by blue-native PAGE (BN-PAGE), similar to what we observed in SDS-PAGE analysis. The complex III-containing supercomplexes also appeared decreased in abundance

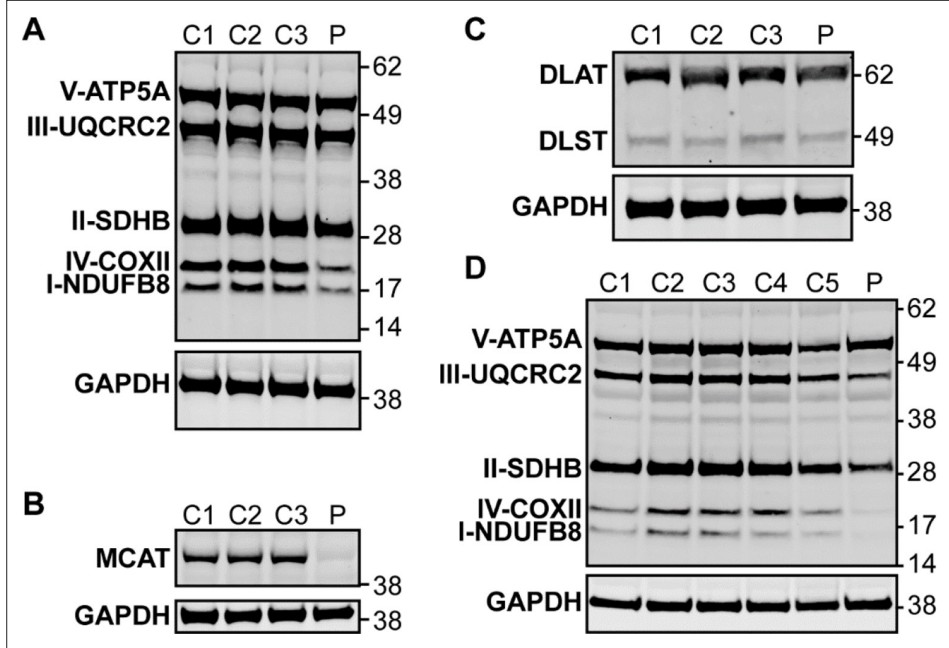

**Figure 2.** Western blot analysis in patient lymphoblasts (**A–C**) and fibroblasts (**D**). (**A**) Western blot analysis to assess the expression level of one subunit of each of the five different oxidative phosphorylation complexes in the patient lymphoblast sample (**P**) compared to three control lymphoblast lines from healthy individuals (C1–3). Protein content of NDUFB8 and COXII, which are subunits of complexes I and IV, respectively, are decreased in the patient compared to healthy controls. (**B**) Western blot analysis to assess malonyl-CoA-acyl carrier protein transacylase (MCAT) levels reveals decreased expression of MCAT in patient lymphoblasts (**P**) compared to three controls (C1–3). (**C**) Western blot analysis to assess lipoylation with an anti-lipoic acid antibody in patient lymphoblasts (**P**) compared to controls (C1–3) reveals normal lipoylation of the PDH and OGDH E2 components (DLAT and DLST, respectively) in the patient sample. (**D**) Western blot analysis to assess the expression level of one subunit of each of the five different oxidative phosphorylation complexes in the patient fibroblast sample (**P**) compared to five fibroblast controls (C1–5). Protein content of NDUFB8, COXII, and SDHB, which are subunits of complexes I, IV, and II, respectively, are decreased in the patient compared to healthy controls.

The online version of this article includes the following source data and figure supplement(s) for figure 2:

**Source data 1.** Uncropped immunoblots for *Figure 2*.

**Source data 2.** Unlabeled immunoblots for *Figure 2*.

**Figure supplement 1.** Quantification of western blot analysis in patient lymphoblasts (**A–C**) and fibroblasts (**D**).

**Figure supplement 1—source data 1.** Excel spreadsheet containing quantitative data for *Figure 2—figure supplement 1*.

**Figure supplement 2.** Measurement of 2-oxoglutaric acid dehydrogenase complex (OGDHc) activity.

**Figure supplement 2—source data 1.** Excel spreadsheet containing quantitative data for *Figure 2—figure supplement 2*.

**Figure supplement 3.** Relative abundance of lactate, 2-oxoglutarate, and 2-oxoadipate from quantitative metabolomics.

**Figure supplement 3—source data 1.** Excel spreadsheet containing quantitative data for *Figure 2—figure supplement 3*.

**Figure supplement 4.** Protein modeling of malonyl-CoA-acyl carrier protein transacylase (MCAT) p.T271I mutation.

in patient fibroblasts, although this can likely be attributed to the observed decreases in complexes I and IV, especially in light of no measured difference in complex III activity. Re-expression of *MCAT* (+hMCAT), but not a mitochondrially targeted GFP construct (+mtGFP), partially rescued these mitochondrial phenotypes in the patient cells (*Figure 3*).

To assess lipoylation, we performed immunoblot analysis in lymphoblasts with an anti-lipoic acid antibody and noted no change in the proband compared to control samples (*Figure 2C*,

**Table 1.** Clinical biochemical testing of respiratory chain enzyme activities in the proband's fibroblasts.

|  | Proband | Reference range |
|---|---|---|
| Complex I | 119 mU/U CS | 163–599 mU/U CS |
| Complex II | 286 mU/U CS | 335–888 mU/U CS |
| Complex III | 632 mU/U CS | 570–1383 mU/U CS |
| Complex IV | 103 mU/U CS | 288–954 mU/U CS |
| Complex V | 630 mU/U CS | 193–819 mU/U CS |
| CS | 412 mU/mg | 151–449 mU/mg |

*Figure 2—figure supplement 1*). We also measured OGDHc activity in lymphoblasts and noted no deficits in the proband compared to controls (*Figure 2—figure supplement 2*). To assess the effects of the homozygous MCAT p.T271I mutation, we also compared metabolite levels between patient fibroblasts and control fibroblast lines. This revealed that lactate is elevated in the patient cells, but metabolites related to other 2-oxoacid dehydrogenase complexes including 2-oxoglutarate and 2-oxoadipate were within the range of normal, consistent with the lack of a deficiency in lipoylation and normal OGDHc activity (*Figure 2—figure supplement 3*).

Although rescue of fully assembled ETC complexes in patient fibroblasts by re-expression of MCAT was incomplete, there are several difficulties associated with this approach, including a lack of isogenic control cell lines and high variability among control cell lines, along with low infection efficiencies in the proband fibroblasts. We therefore decided to test whether human MCAT with the p.T271I mutation could efficiently restore protein lipoylation, ETC complex assembly, and cellular respiration in our previously generated hypomorphic *Mcat* mutant C2C12 cell lines (*Nowinski et al., 2020*). These cells display a roughly 90% reduction in MCAT expression, and similar to the proband fibroblasts have defects in assembly and stability of ETC complexes I, II, and IV. However, the hypomorphic *Mcat* C2C12 lines also have undetectable protein lipoylation by western blot (*Nowinski et al., 2020*). Re-expression of wild-type and p.T271I human MCAT rescued protein lipoylation (*Figure 4A*). The effect on the expression of NDUFA9, a complex I subunit, was more variable (*Figure 4A*). We therefore assessed how re-expression of wild-type and p.T271I human MCAT affects mitochondrial respiration. In comparison to controls, *Mcat* hypomorphic mutant C2C12 exhibited significantly decreased basal and maximal respiration rates, as previously described (*Figure 4B and C*; *Nowinski et al., 2020*). Expression of a mitochondrially targeted DSRed construct (mtDSRed) minimally affected respiration in the *Mcat* mutant cells, while expression of wild-type human MCAT fully rescued both basal and maximal respiration rates. In contrast, expression of the p.T271I mutant only partially rescued respiration (*Figure 4B and C*).

## Discussion

Here, we present the first case of a patient with a biallelic variant in *MCAT* with combined oxidative phosphorylation deficiency. The identified mutation, c.812T>C; p.T271I, is located adjacent to the active site, and likely affects both the stability and catalytic activity of the enzyme (*Figure 2— figure supplement 4*). Defects in complexes I, II, and IV were noted in fibroblasts based on enzyme activity assays and immunoblotting for individual subunits of the complexes. Additionally, subunits of complexes I and IV were also decreased in lymphoblasts. BN-PAGE revealed that intact and fully assembled supercomplexes I, III, and IV were decreased in abundance in patient fibroblasts. Wild-type *MCAT* led to increased abundance of each of these three complexes proving causality of the *MCAT* variant. Taken together, we demonstrate that the homozygous variant in *MCAT* c.812T>C; p.T271I leads to combined oxidative phosphorylation deficiency.

Interestingly, although *MCAT* plays a key role in the mtFAS system, which generates octanoate, the precursor for lipoate, our patient did not have a lipoylation defect noted in lymphoblasts and had normal OGDHc activity in lymphoblasts (*Mayr et al., 2014*). Consistently, other biochemical findings that would indicate a lipoate deficiency such as lactic acidemia, elevated urine 2-oxoglutarate, and

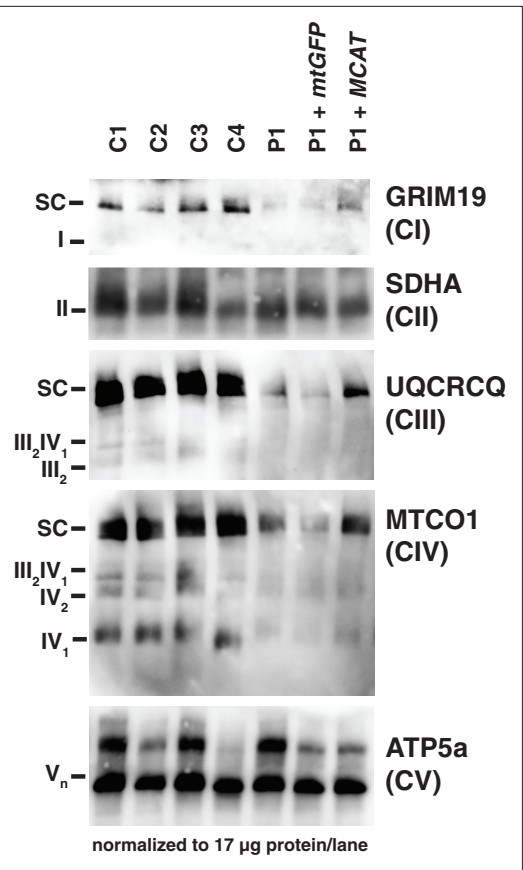

**Figure 3.** Mitochondrial respiratory supercomplexes visualized by blue-native PAGE. Mitochondrial lysates generated from four control fibroblast lines (C1–4), the patient fibroblast line (**P**), the patient line expressing the control plasmid mtGFP (*P*+mtGFP), and the patient line with re-expression of wild-type malonyl-CoA-acyl carrier protein transacylase (MCAT) (*P*+MCAT). The lysates are separated by blue-native PAGE and immunoblotted with the indicated antibodies. Expression of GRIM19, UQCRCQ, and MTCO1 delineating supercomplexes I, III, and IV respectively are decreased in the patient cells (**P**) compared to four controls, and improved with re-expression of wild-type MCAT (*P*+MCAT). For each immunoblot, at least two technical replicates were completed and one representative blot is shown. SC = supercomplex.

The online version of this article includes the following source data for figure 3:

**Source data 1.** Uncropped immunoblots for *Figure 3*.

**Source data 2.** Unlabeled immunoblots for *Figure 3*.

elevated blood glycine were absent. This finding is unique relative to other, previously described pathogenic mtFAS mutations, which usually display both lipoylation and ETC defects (*Heimer et al., 2016*; *Liu et al., 2021*). We hypothesize that this decoupling of mtFAS endpoints may be a result of substrate limitation. Because MCAT supplies malonyl-ACP which is necessary to extend acyl chains, this less functional MCAT mutant may decrease the available malonyl-ACP and therefore preference the pathway toward shorter chain products (like lipoic acid) over the longer acyl chains that are required for LYRM binding and ETC assembly.

Recently, double homozygous mutations in *MCAT* (p.L81R and p.R212W) were identified in a consanguineous Chinese family with isolated, progressive autosomal recessive optic neuropathy. These patients were otherwise healthy, had normal motor function, and brain MR imaging revealed only bilateral thinning of the optic nerve. Overexpression of L81R MCAT or double mutant L81R/R212W MCAT led to decreased COX4 expression in the mitochondria in HEK293T cells. The authors hypothesize that MCAT L81R and R212W variants only affect retinal ganglion cell axon maintenance (*Li et al., 2020*). Our patient with the homozygous MCAT p.T271I pathogenic variant exhibits more severe mitochondrial disease characterized by multi-system involvement.

## Materials and methods
### Human subjects enrollment

The family was enrolled in Icahn School of Medicine at Mount Sinai (13-00495; 17-02143) and University of Wisconsin School of Medicine and Public Health (2021-0723) approved genetics research studies. Written informed consent was obtained, and all investigations were conducted in accordance with the principles of the Declaration of Helsinki.

### Whole exome sequencing

Raw whole exome sequencing data (fastq files) for the proband and both parents (trio testing) were received from an outside clinical testing laboratory and were reanalyzed. Sequence reads were independently mapped to the reference genome (hg19) with BWA-MEM and further processed using the GATK Best Practices workflows (*Li and Durbin, 2009*; *Van der Auwera et al., 2013*; *McKenna et al., 2010*); 96% of the exome was covered at ≥10×. Single nucleotide variants and small indels were called with GATK HaplotypeCaller and annotated using ANNOVAR, dbSNP (v138), 1000 Genomes (August 2015), NHLBI Exome Variant Server (EVS), and ExAC (v3) (*Wang et al., 2010*; *Sherry et al., 2001*; *Genomes Project et al., 2015*; *Lek et al., 2016*). Called variants were filtered

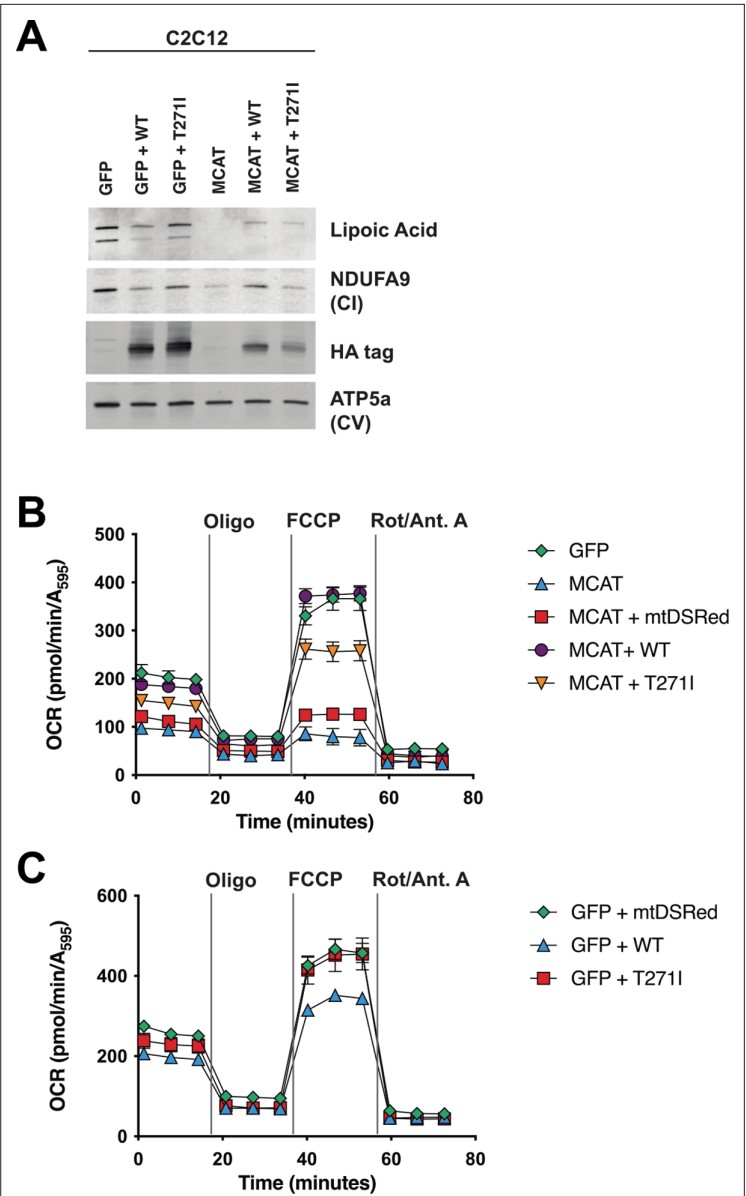

**Figure 4.** Rescue of hypomorphic malonyl-CoA-acyl carrier protein transacylase (MCAT) mutant C2C12 with wild-type (WT) and p.T271I mutant MCAT constructs. (**A**) Mitochondrial lysates generated from MCAT hypomorphic CRISPR mutant C2C12 mouse skeletal myoblasts (MCAT) and isogenic controls (GFP), stably infected with either WT human MCAT or p.T271I mutant human MCAT (T271I) transgenes. Lysates were separated by SDS-PAGE and immunoblotted with the indicated antibodies. Protein lipoylation (lipoic acid) is undetectable in MCAT mutant C2C12 and rescued by both WT and p.T271I mutant MCAT, while complex I (NDUFA9, CI) is only rescued by expression of WT human MCAT. For each immunoblot, at least two technical replicates were completed and one representative blot is shown. (**B–C**) Cells from each of the indicated genotypes were seeded in eight wells of a 96-well seahorse plate and allowed to adhere overnight, then equilibrated and treated with the indicated drugs following standard mitochondrial stress test protocols from the manufacturer to determine oxygen consumption rate (OCR). Error bars are SEM. Oxygen consumption is fully rescued by WT human MCAT but only partially rescued by expression of the p.T271I patient mutation.

The online version of this article includes the following source data for figure 4:

**Source data 1.** Uncropped immunoblots for *Figure 4*.

**Source data 2.** Unlabeled immunoblots for *Figure 4*.

**Source data 3.** Excel spreadsheet containing quantitative data for *Figure 4* Panel B.

**Source data 4.** Excel spreadsheet containing quantitative data for *Figure 4* Panel C.

with Ingenuity Variant Analysis (Qiagen, Redwood City, CA, USA, http://www.ingenuity.com). A total of 102,052 total variants in 17,924 genes were identified in the three samples. Variants were filtered based on confidence (call quality of ≥20, passed upstream pipeline filtering, and outside top 3% most exonically variable 100 base windows and/or 3% most exonically variable genes in healthy public genomes included), frequency (variants excluded if frequency was at least 0.2% in the 1000 Genomes Project, NHLBI ESP exomes, or ExAC), predicted deleteriousness (frameshift, in-frame indel, start/stop codon changes, missense changes, splice site loss up to six bases into intron or as predicted by MaxEntScan, and variants listed in HGMD were included), and genetic analysis (de novo or recessive inheritance considered). This filtering strategy resulted in the identification of 21 variants in 20 genes. All of these variants were considered, and knowledge of gene function was reviewed.

## Cell lines

A blood sample was collected from the male proband and a lymphoblast line was set up according to standard practices. A skin biopsy was obtained from the proband and a fibroblast culture was established according to standard practices. All patient and control human cell lines receive authentication by genotyping and chromosomal microarray prior to freezing large stocks. All cell lines are tested monthly for mycoplasma contamination.

## Oxidative phosphorylation enzyme activities

The enzyme activities of complexes I, II, III, IV, and V and citrate synthase in fibroblasts were measured using spectrophotometry in a mitochondrial enriched fraction (*Rodenburg, 2011*).

## Immunoblot analysis

Lymphoblast and fibroblast samples from the proband and control samples were utilized for immunoblot analysis and processed as described (*Houten et al., 2014*). The membrane was probed with total OXPHOS human WB antibody cocktail at 1:1000 (MitoSciences, ab110411), anti-MCAT antibody at 1:1000 (Sigma-Aldrich, HPA035471, RRID:AB_10670590), anti-lipoic acid antibody at 1:1000 (Calbiochem, 437695), anti-DLAT at 1:1000 (Abcam, ab172617, RRID:AB_2827534), and/or anti-DLST at 1:1000 (Cell Signaling, 5556, RRID:AB_106951) with anti-GAPDH antibody at 1:2500 (Abnova, H00002597-K) used as a loading control. For each immunoblot, at least two technical replicates were completed and one representative blot is shown and quantified.

## OGDHc activity

Control and patient-derived lymphoblasts were resuspended in PBS, lysed by sonication and centrifuged for 10 min at $1000 \times g$ at 4°C. The supernatant (60 µg total protein) was mixed with assay buffer (final concentration: 50 mM MOPS, pH 7.4, 0.2 mM $MgCl_2$, 0.01 mM $CaCl_2$, 0.3 mM TPP (C8754, Sigma), 0.12 mM CoA (C3144, Sigma), 2 mM $NAD^+$ (N6522, Sigma), 2.6 mM β-mercaptoethanol). The reaction was started by the addition of 1 mM 2-oxoglutarate substrate (K1128, Sigma). The activity of the OGDHc was followed by measuring the NADH production at 340 nm at 30°C and steady-state velocities were calculated from the linear portion of the time curve.

## Rescue of proband fibroblasts

For rescue experiments, proband fibroblasts were infected with lentivirus harboring a control transfer plasmid (pLenti mtGFP) or pLenti expressing *MCAT* off the full CMV promoter (pLenti hMCAT). Human *MCAT* was cloned from a mixture of human cDNA isolated from the common cell lines HCT116, HCT15, and 293T. Virus was packaged in HEK293T cells, collected at 48–72 hr post-transfection, and viral supernatants were filtered and applied directly to patient fibroblasts for 16 hr. Media was changed and replaced with fresh media and cells were allowed to expand and recover for 48 hr, then harvested for BN-PAGE.

## Crude mitochondrial isolation and BN-PAGE

Cells were harvested with 0.25% trypsin-EDTA (Gibco, 25200-072), pelleted, washed with sterile PBS (Gibco, 10010-023), pelleted again, and frozen in liquid nitrogen. Mitochondria were isolated as previously described (*Nowinski et al., 2020*). Briefly, cells were suspended in 1 mL CP-1 buffer

(100 mM KCl, 50 mM Tris-HCl, 2 mM EGTA, pH 7.4), lysed by passing through a 27-gauge needle, and centrifuged at 700 × $g$ to pellet debris. Supernatant was subsequently centrifuged at 10,000 × $g$ to pellet crude mitochondria. Mitochondrial pellets were resuspended in CP-1 buffer, normalized for mitochondrial protein by BCA Assay (Pierce), and used in assays described below. BN-PAGE was performed using the Invitrogen NativePAGE system. One hundred micrograms of mitochondria were resuspended in 1× pink lysis buffer (Invitrogen, BN20032). Digitonin (GoldBio D-180-2.5) was added to a final concentration of 1% mass/volume. Samples were incubated on ice for 15 min, then spun for 20 min at 20,000 × $g$. Six microliters of NativePAGE sample buffer (Invitrogen, BN20041) was added, and 10 µL of sample was run on precast 3–12% NativePAGE gels (Invitrogen, BN2011BX10) according to the manufacturer's instructions. Gels were subsequently transferred to nitrocellulose and blotted with the indicated primary antibodies. Secondary anti-mouse HRP antibody and SuperSignal West Femto Maximum Sensitivity Substrate (Thermo Fisher Scientific, 34096) was used to visualize bands on film (GeneMate, F-9024-8X10).

## Metabolomics analysis

Cells for metabolomics analysis were grown simultaneously with samples for BN-PAGE assessment. One hundred thousand cells were seeded and then collected after 48 hr. Cells were collected and lysed in 80:20 optima-grade methanol:water and frozen in liquid nitrogen. After three consecutive freeze-thaw cycles, samples were centrifuged at top speed for 10 min in a refrigerated centrifuge. Supernatants were transferred to fresh tubes and a protein assay was performed on the methanol lysate. Metabolite extracts were dried in a sample concentrator and then metabolites were resuspended in 80:20 acetonitrile:water relative to their protein abundance in preparation for LCMS analysis. Data acquisition was performed by reverse-phase chromatography on a 1290 UHPLC liquid chromatography (LC) system interfaced to a high-resolution mass spectrometry 6550 iFunnel Q-TOF mass spectrometer (MS) (Agilent Technologies, CA, USA). The MS was operated in both positive and negative (ESI+ and ESI-) modes. Analytes were separated on a Sequant ZIC-pHILIC column (5 µm, 2.1×150 mm, Merck, Damstadt, Germany). The column was kept at room temperature. Mobile phase A composition was 10 mM ammonium acetate in water (pH 9.8) and mobile phase B composition was 100% acetonitrile. The LC gradient was 0 min: 90% B; 15 min: 30% B; 18 min: 30% B; 19 min: 10% B; and 27 min: 10% B. The flow rate was 250 µL/min. The sample injection volume was 10 µL. ESI source conditions were set as follows: dry gas temperature 225°C and flow 18 L/ min, fragmentor voltage 175 V, sheath gas temperature 350°C, and flow 12 L/min, nozzle voltage 500 V, and capillary voltage +3500 V in positive mode and −3500 V in negative. The instrument was set to acquire over the full m/z range of 40–1700 in both modes, with the MS acquisition rate of 1 spectrum/s in profile format. Raw data files (.d) were processed using Profinder B.08.00 SP3 software (Agilent Technologies, CA, USA) with an in-house database containing retention time and accurate mass information on 600 standards from Mass Spectrometry Metabolite Library (IROA Technologies, MA, USA) which was created under the same analysis conditions. The in-house database matching parameters were: mass tolerance 10 ppm; retention time tolerance 0.5 min. Peak integration result was manually curated in Profinder for improved consistency and exported as a spreadsheet (.csv). Peak areas were blank subtracted and normalized to total ion content. The experiment was replicated with consistent results.

## Rescue of MCAT CRISPR mutant C2C12 cell lines

Two clonal C2C12 cell lines with mutations in *MCAT* were generated and thoroughly characterized previously (*Nowinski et al., 2020*). These cell lines or a control clonal line (GFP) were infected with retrovirus expressing the human MCAT CDS (WT), human MCAT harboring the T271I mutation (T271I), or a mitochondrially targeted DSRed control (mtDSRed). Stably infected cells were selected by culturing in normal growth medium (DMEM [Corning 10-013-CV]+10% FBS) supplemented with 2 µg/mL puromycin for 7 days. Cells were expanded and harvested for mitochondrial isolation and western blotting at 70% confluency or plated for Seahorse assays. For immunoblotting, samples were processed as published previously (*Nowinski et al., 2020*).

## Seahorse assay for mitochondrial respiration

Cells were plated in 8 wells of a 96-well Seahorse plate at 6000 cells/well in DMEM+10% FBS and incubated overnight. Prior to the assay, cells were washed 2× and media was replaced with XF base DMEM (Agilent 103334-100) supplemented with 25 mM glucose (Agilent 103577-100), 2 mM glutamine (Agilent 103579-100), and 1 mM pyruvate (Agilent 103578-100), pH 7.4. A mitochondrial stress test was performed on a Seahorse XFe96 Analyser under standard assay conditions according to the manufacturer's instructions using 1 µM oligomycin, 3 µM FCCP, and 0.5 µM Rotenone+0.5 µM Antimycin A (Agilent 103015-100). Data were normalized to total cell content per well using crystal violet staining. Briefly, cells were fixed in 4% paraformaldehyde, washed, stained with crystal violet. Excess dye was washed away, and then cells were permeabilized using 1% SDS to solubilize the crystal violet stain. Samples were transferred to a standard 96-well plate and absorbance at 595 nm was read on a BioTek plate reader. Results were analyzed in WAVE software and processed through the XF Mito Stress Test Report. The experiment was replicated with consistent results.

## Modeling of MCAT p.T271I mutation

The human MCAT crystal structure was downloaded from the Protein Data Bank (PDB 2C2N). The crystal sequence was aligned with the full-length sequence (Uniprot: Q8IVS2) to identify the residue in the crystal structure (T235) that corresponds with the mutated residue identified in the patient (T271). The crystal structure was imported into PyMol 2.5.2, and T235 was mutated to an isoleucine in order to model the structural impact of the mutation on interactions with nearby residues.

## Acknowledgements

We are grateful to the patient and family for participating in this study. We are thankful to Emalyn Cork, MS, CGC, who assisted in the care of this patient. The funders had no role in study design, data collection and interpretation, or the decision to submit the work for publication.

## Additional information

### Funding

| Funder | Grant reference number | Author |
|---|---|---|
| Eunice Kennedy Shriver National Institute of Child Health and Human Development | K08HD086827 | Bryn D Webb |
| National Institute of Diabetes and Digestive and Kidney Diseases | R01DK113172 | Sander M Houten |
| National Institute of General Medical Sciences | GM115174 | Jared Rutter |
| The Nora Eccles Treadwell Foundation | | Jared Rutter |
| Howard Hughes Medical Institute | | Ralph J DeBerardinis |
| National Cancer Institute | | Ralph J DeBerardinis |
| The Once Upon a Time Foundation | | Ralph J DeBerardinis |
| National Heart, Lung, and Blood Institute | T32HL007576 | Sara M Nowinski |
| National Institute of General Medical Sciences | GM115129 | Jared Rutter |

| Funder | Grant reference number | Author |
|---|---|---|
| Eunice Kennedy Shriver National Institute of Child Health and Human Development | F32HD096786 | Ashley Solmonson |
| National Institute of General Medical Sciences | GM110755 | Sara M Nowinski |

The funders had no role in study design, data collection and interpretation, or the decision to submit the work for publication.

## Author contributions

Bryn D Webb, Conceptualization, Data curation, Formal analysis, Funding acquisition, Validation, Investigation, Visualization, Methodology, Writing – original draft, Writing – review and editing; Sara M Nowinski, Ashley Solmonson, Formal analysis, Investigation, Visualization, Methodology, Writing – review and editing; Jaya Ganesh, Data curation, Writing – review and editing; Richard J Rodenburg, Joao Leandro, Hieu S Vu, Investigation; Anthony Evans, Investigation, Visualization; Thomas P Naidich, Data curation, Formal analysis, Writing – review and editing; Bruce D Gelb, Supervision, Writing – review and editing; Ralph J DeBerardinis, Jared Rutter, Supervision, Methodology, Writing – review and editing; Sander M Houten, Conceptualization, Supervision, Investigation, Methodology, Writing – review and editing

## Author ORCIDs

Bryn D Webb http://orcid.org/0000-0001-6174-4677
Ashley Solmonson http://orcid.org/0000-0001-8863-4558
Jared Rutter http://orcid.org/0000-0002-2710-9765

## Ethics

Informed consent and consent to publish was obtained from the index family. The family was enrolled into IRB-approved studies at the Icahn School of Medicine at Mount Sinai (17-02143; 13-00495) and the University of Wisconsin School of Medicine and Public Health (2021-0723).

## Decision letter and Author response

Decision letter https://doi.org/10.7554/eLife.68047.sa1
Author response https://doi.org/10.7554/eLife.68047.sa2

# Additional files

## Supplementary files
• MDAR checklist

## Data availability

This is a rare disorder with only a single case worldwide currently known and thus there are privacy concerns. In alignment with the family we prefer to not make the exome data publicly available. However, we are able to share a variant call file (vcf) with interested investigators without further restrictions. The investigator should contact the corresponding author with this request. The references for the pipeline to analyze sequencing data are provided. No custom code was used in this analysis. The raw data for generation of all graphs has been included as source data files.

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

## Appendix 1

**Appendix 1—key resources table**

| Reagent type (species) or resource | Designation | Source or reference | Identifiers | Additional information |
|---|---|---|---|---|
| Cell line (*Mus musculus*) | C2C12 | ATCC | #CRL-1772, RRID:CVCL_0188 | |
| Cell line (*Homo sapiens*) | HEK293T | ATCC | #CRL-11268, RRID:CVCL_1926 | |
| Antibody | Anti-human OXPHOS cocktail (mouse monoclonal) | MitoSciences | ab110411 | WB: (1:1000) |
| Antibody | Anti-GAPDH (rabbit monoclonal) | Abnova | H00002597-K | WB: (1:2500) |
| Antibody | Anti-MCAT (mouse monoclonal) | Sigma-Aldrich | HPA035471, RRID: AB_10670590 | WB: (1:1000) |
| Antibody | Anti-MCAT (mouse monoclonal) | Santa Cruz | sc-390858, RRID:AB_2827536 | WB: (1:100) |
| Antibody | Anti-DLAT (rabbit monoclonal) | Abcam | ab172617, RRID:AB_2827534 | WB: (1:1000) |
| Antibody | Anti-DLST (rabbit polyclonal) | Cell Signaling | 5556, RRID:AB_106951 | WB: (1:1000) |
| Antibody | Anti-GRIM19 (mouse monoclonal) | Abcam | ab110240, RRID:AB_10863178 | WB: (1:1000) |
| Antibody | Anti-SDHA (mouse monoclonal) | Abcam | ab14715, RRID:AB_301433 | WB: (1:10,000) |
| Antibody | Anti-UQCRQ (mouse monoclonal) | Abcam | ab110255 | WB (1:1000) |
| Antibody | Anti-MTCO1 (mouse monoclonal) | Abcam | ab14705, RRID:AB_2084810 | WB (1:1000) |
| Antibody | Anti-ATP5A (mouse monoclonal) | Abcam | ab14748, RRID:AB_301447 | WB (1:1000) |
| Antibody | Anti-Lipoic Acid (rabbit polyclonal) | Abcam | ab58724, RRID:AB_880635 | WB (1:1000) |
| Antibody | Anti-NDUFA9 (mouse monoclonal) | Abcam | ab14713, RRID:AB_301431 | WB (1:1000) |
| Antibody | Anti-HA (rabbit polyclonal) | BioLegend | PRB-101C | WB (1:1000) |

