## [Editor Report]

This paper is a valuable report of a human subject with an MCAT mutation showing reduced mitochondrial activity. Analysis of cells from this subject convincingly showed similar abnormalities as previously demonstrated for MCAT deficiency in the mouse. This work adds to the field of mitochondrial disease genes and their impact on human physiology.

---

## [Decision Letter]

**Decision letter after peer review:**

Thank you for submitting your article "Recessive pathogenic variants in MCAT cause combined oxidative phosphorylation deficiency" for consideration by *eLife*. Your article has been reviewed by 2 peer reviewers, one of whom is a member of our Board of Reviewing Editors, and the evaluation has been overseen by Utpal Banerjee as the Senior Editor. The reviewers have opted to remain anonymous.

The reviewers have discussed their reviews with one another, and the Reviewing Editor has drafted this to help you prepare a revised submission. As you can tell from the comments below, the reviewers believe manuscript is of interest and adds further translational value to the previous work of the authors, but issues remain unresolved. In general, the study would benefit from more controlled reconstitution assays (wild-type and T271I-MCAT) in a defined in vitro cell model, such as the previously generated hypomorphic Mcat clones.

Essential revisions:

1) Some data on the molecular mechanism connecting the loss of MCAT activity to mitochondrial dysfunction is needed. In the absence of such data, a specialty journal may be more appropriate.

2) The amino acid substitution (T271I) is in the vicinity of the predicted active site (His270) of the MCAT enzyme: how does this mutation impact catalytic activity and protein stability?

3) How do the authors reconcile this defect in MCAT with no apparent lesions in the mtFAS system. A more thorough analysis of the effects of this T271I-MCAT variant on mitochondrial functionality (e.g., by using Seahorse respirometry) would complement the presented data.

4) Quantification of the Western blots in Figure 2 and statistical significance analysis is needed.

---

## [Author Response]

Essential revisions:1) Some data on the molecular mechanism connecting the loss of MCAT activity to mitochondrial dysfunction is needed. In the absence of such data, a specialty journal may be more appropriate.

To provide additional clarity and mechanistic data in an isogenic cell system with better controls, we created new C2C12 cell lines that express wild-type human MCAT or the T271I mutant in both control C2C12 and our Mcat hypomorphic CRIPSR mutants described in detail in our previous *eLife* publication. Using total protein lysates from these new C2C12 lines, we performed western blots and found that the T271I mutant rescues protein lipoylation in MCAT-deficient cells to a similar extent as does wild-type human MCAT. In contrast, respiration was not rescued by the T271I mutant. We previously showed that the electron transfer chain (ETC) assembly function of mtFAS is mediated by LYRM proteins, which are destabilized when mtFAS is not functional. We added a new section to the discussion that describes the potential involvement of LYRM proteins.

2) The amino acid substitution (T271I) is in the vicinity of the predicted active site (His270) of the MCAT enzyme: how does this mutation impact catalytic activity and protein stability?

MCAT abundance is decreased in the patient cells (Figure 2), implying decreased protein stability. We also addressed this question in C2C12 cells by expressing wild-type and T271I-mutant MCAT, driven by the EF1-COR promoter (Figure 4A), and found that the mutant is less stable. We also include a model of the mutation (Figure 2—figure supplement 2D) showing its proximity to the active site. Based on these data, we believe that both stability and catalytic activity are likely affected and have added a section to the Discussion considering this.

3) How do the authors reconcile this defect in MCAT with no apparent lesions in the mtFAS system. A more thorough analysis of the effects of this T271I-MCAT variant on mitochondrial functionality (e.g., by using Seahorse respirometry) would complement the presented data.

We believe that this mutation in MCAT is unique in that it seems to decouple two endpoints downstream of mtFAS: lipoic acid synthesis, which is preserved, and LYRM stability and ETC assembly, which we previously showed does not depend on lipoic acid synthesis. We have performed Seahorse respirometry in the newly created C2C12 cell lines and find that the T271I mutant only partially rescues both basal and maximal respiration, in comparison to wild-type human MCAT, which rescues to control levels. These data agree well with our western blot data that show that lipoylation is rescued to a similar extent by both constructs but ETC assembly is still impaired.

We hypothesize that this decoupling may be a result of substrate limitation: because MCAT supplies malonyl- ACP, which is necessary to extend acyl chains, this less functional MCAT mutant may decrease the available malonyl-ACP and, therefore, preference the pathway towards shorter chain products (like lipoic acid) over the longer acyl chains required for LYRM binding and ETC assembly. We have added a section proposing this hypothesis in the Discussion.

4) Quantification of the Western blots in Figure 2 and statistical significance analysis is needed.

We quantified the Western blots and added descriptive statistics in Figure 2 (Figure 2—figure supplement 2A).